# Rhizosphere-Associated Microbiome Profile of Agricultural Reclaimed Lands in Egypt

Mohamed Hassan Korkar, Mahmoud Magdy *, Samah Mohamed Rizk, Yosur Gamal Fiteha, Aiman Hanafy Atta and Mohamed Abdel-Salam Rashed

Genetics Department, Faculty of Agriculture, Ain Shams University, Cairo 11241, Egypt
* Correspondence: m.elmosallamy@agr.asu.edu.eg; Tel.: +20-1093411288

**Abstract:** Plants especially in their natural habitat are considered part of a rich ecosystem that includes many various microorganisms in the soil. The current study aimed to identify the bacterial profile of agriculture-related soil samples using the metabarcoding technique to compare and explore relevant rhizosphere bacteria associated with plant cultivations in newly reclaimed land and habitual cultivated ones. The total environmental DNA was extracted from rhizosphere and noncultivated samples derived from three land types in Egypt. The bacterial 16S rDNA was amplified and sequenced by NGS technology to profile each sample. The microbial profile was characterized by statistical and literature-based methods. Among all samples, the most identified phyla were Actinobacteriota (28%), followed by Proteobacteriota (26%), Firmicutes (14%), Acidobacteriota and Chloroflexi (7%), Gemmatimonadota (5%), Bacteriodota and Crenarchaeota (3%), and Myxococcota (2%), in addition to 37 other phyla with <1% counts. A total of 74 OTUs were unique to the plant rhizosphere area and classified as Bacteriodota (5.1%:0.3%), Firmicutes (2.4%:0.1%), and Proteobacteria (3.5%:2%) phyla in agricultural and reclaimed lands, respectively. Moreover, the rhizosphere profile included a large portion of uncultured and unidentified bacterial species, which opened a window to further analysis. Our analysis provides a key knowledge about the rhizosphere microbiome and highlights its possible use to create microbial-based biofertilizers targeting plant performance in contrast to traditional fertilizers and their side effect on the agriculture sector.

**Keywords:** 16S rDNA metabarcoding; microbial community; newly reclaimed lands; exophyta; soil microbiota

## 1. Introduction

Plants, especially in their natural habitat, are considered part of a rich ecosystem that includes many various microorganisms in the soil [1], in addition to microbes, which are functioning as a community that forms ecological niches [2]. Soil is the ultimate reservoir of culturable and nonculturable microorganisms and provides different environments and nutrients for their survival [3], where microbial populations are instrumental to fundamental processes that drive the stability and productivity of agroecosystems [4]. Soil microorganisms (i.e., bacteria, fungi, actinomycetes, and total microorganisms), help in the safe growth of plants, in addition to microbes [3], and are correlated with soil physicochemical properties (soil pH, soil moisture, soil temperature, soil carbon, and nitrogen contents) [5]. Microorganisms play a dichotomous role in the soil nitrogen cycle through mineralization and immobilization, therefore contributing to the maintenance of soil fertility and mitigation of global warming [5]. They catalyze various biochemical processes (decomposition, nutrient turnover, and degradation of pesticides and toxic metabolites of the soil) and can be used to assess soil quality and health [3]. In the last few years, great progress has been made in the knowledge of the composition of rhizosphere microbiomes and their dynamics [1]. The rhizosphere microbiome plays a key role in plant nutrient provision [6]. Rhizosphere microorganisms offer to host plants the essential assimilable

nutrients, stimulate the growth and development of host plants, and induce antibiotics production [7]. Recent advances in microbe–plant interactions research have revealed that plants have the ability to shape their rhizosphere microbiome, as proved by the fact that specific microbial communities are hosted in different plant species even when they grow on the same soil [8], and the growth of rhizosphere micro-organisms is regulated by the phytoproducts excreted from plant roots; this excretion of phytochemicals alters the chemistry of rhizosphere soil and also commands the fate of linked organisms and vice versa [2].

Beneficial microbes in the microbiome of plant roots improve plant health. Induced systemic resistance (ISR) emerges as an important tool by which selected plant-growth-promoting bacteria (PGPB) in the rhizosphere stimulates the plant for improved defense versus a wide range of insect herbivores and pathogens; for example, two genera, Azospirillum and Azotobacter, were found in abundance in soil [9]. A wide variety of root-associated mutualists, including Pseudomonas, Bacillus, Trichoderma, and Mycorrhiza species, sensitize the plant immune system to enhance its defense without directly activating costly defenses [9].

Using standard methods for DNA extraction often results in poor quality and low yield making them inappropriate for the analysis of community through polymerase chain reaction (PCR) because of the presence of humic substances and the formation of chimeric products with smaller template DNAs [10], so the requirement for the development of the metagenomic information of provincially important crops and their plant interactions with microorganisms and agricultural performers, for narrowing down important information from huge databases, have been recommended. The role of a functional and taxonomical diversity of soil microorganisms in understanding soil repression and the portion played by the microbial metabolites in the process have been evaluated and discussed in the context of the "omics" approach. "Omics" studies have discovered significant data about microbial variety, their responses to numerous biotic and abiotic stimuli, in addition to the physiology of disease repression [11]. Currently, NGS methods such as 16S rDNA/rRNA metagenomics or amplicon sequencing are intended for the taxonomic profiling of the soil microbial communities. Although 16S rDNA/rRNA NGS-based microbial data are not suited for the analysis of the functional potential of the recognized operational taxonomic units as compared to shotgun metagenomics, current developments in the bioinformatics discipline allow the performance of such studies [12]. Advances in next-generation sequencing (NGS) platforms, gene editing technologies, metagenomics, and bioinformatics approaches allow us to unravel the entangled webs of interactions of holobionts and core microbiomes for efficiently deploying the microbiome to increase crops' nutrient acquisition and resistance to abiotic and biotic stresses [13], in addition to the microbial diversity analysis from these environments that will help identify new microorganisms having specificity for unique applications [14].

The current study aimed to profile the bacterial community of agriculture-related soil samples using the metabarcoding technique. The study was designed to explore and compare the relevant rhizosphere bacteria associated with plant cultivations by contrasting three land types (newly reclaimed lands and habitual cultivated ones, controlled by uncultivated desert lands). The complete metabarcoding profiling tests whether the bacteria inhabiting the rhizosphere area of cultivated plants are associated with the agriculture practice regardless of the soil type or the microbial source, and eventually recommend the identified beneficial bacteria as biofertilizers in any land type.

## 2. Materials and Methods

### 2.1. Sampling Locations

Field sampling used in this study was conducted in February 2021. Fifteen soil samples were collected from three different locations in Egypt, namely, agricultural land in Qalubiya governorate (30°7′42″ N and 31°14′32″ E), horticulture farm on Cairo-Alexandria Road (31°12′20″ N and 29°55′28″ E), and a desert land (29°20′32″ N and 25°5′19″ E). For

agricultural soils and reclaimed lands, three subplots covered by different plantation types were considered (Table 1). Plants with the same growth status were randomly selected from similar subplots. The soil of the same sampling point was mixed to obtain a composite sample. Within each land type, for each plant species, triplicate composite soil samples were taken from the root rhizosphere area (i.e., soils adjacent to the plant roots at ~5 cm depth) and the area between two plants of the same type (Figure 1). The replicates were approximately 2 m apart from each other).

**Table 1.** A list of the soil sample collection describing the sampled lands type, plantations type, samples type (rhizosphere and uncultivated areas), and sample code.

| Land | Plantation | Sample | Code |
|---|---|---|---|
| Agricultural Lands (N) | Citrus trees | Rhizosphere area | NP1 |
| | | Uncultivated area | NC1 |
| | Olives trees | Rhizosphere area | NP2 |
| | | Uncultivated area | NC2 |
| | Fava beans | Rhizosphere area | NP3 |
| | | Uncultivated area | NC3 |
| Reclaimed Lands (R) | Apricot trees | Rhizosphere area | RP1 |
| | | Uncultivated area | RC1 |
| | Pear trees | Rhizosphere area | RP2 |
| | | Uncultivated area | RC2 |
| | Eggplants | Rhizosphere area | RP3 |
| | | Uncultivated area | RC3 |
| Desert Lands (D) | No plantations | Uncultivated area | DC1 |
| | | Uncultivated area | DC2 |
| | | Uncultivated area | DC3 |

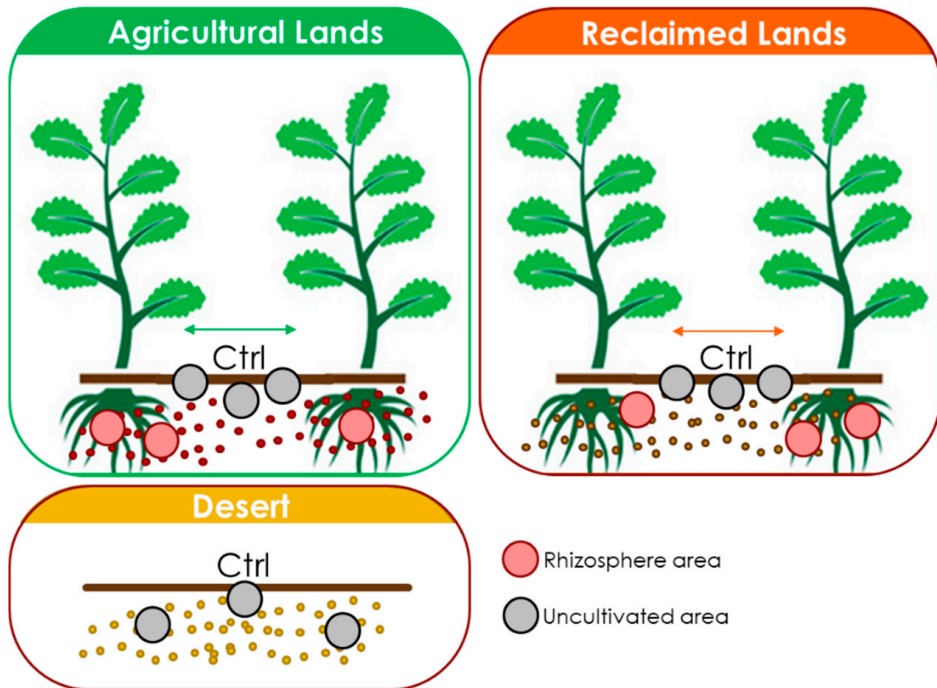

**Figure 1.** A schematic representation of the sampling process from three land types (agricultural, reclaimed, and desert), at each land type samples around plant roots (i.e., rhizosphere) and from the uncultivated area at the same depth. Samples were collected in triplicates.

## 2.2. Sampling Technique and Soil Physicochemical Characterization

Samples of soil were taken with a sterile spatula into Falcon™ 50 mL sterile plastic tubes from the surface layer (0–15 cm depth), labeled with a site code, and kept at room temperature until examination. The samples were divided into two subsamples: one was air-dried and then stored at 25 °C to determine the chemical properties, and the other was stored at −80 °C for DNA extraction. The organic matter was measured by titration using the standard laboratory protocols for soil samples [15,16]. Elemental concentrations (mg $L^{-1}$) were measured using atomic emission spectrometry (AES), atomic absorption spectrometry (AAS), and inductively coupled plasma atomic emission spectrometry (ICP-AES) following Gavlak et al. [17].

## 2.3. DNA Extraction

Genomic DNA was extracted from soil microorganisms via chemical and mechanical lysis using the Power Soil MoBio DNA Isolation Kit (MO BIO Laboratories Inc., Carlsbad, CA, USA), according to the manufacturer's instructions, with a final elution volume of 100 mL. Subsequently, the DNA extract was checked on 1% agarose gel, and DNA concentration and purity were determined with a Quantus™ Fluorometer (Promega, Madison, WI, USA). Extracted DNA was stored at −20 °C until required for PCR.

## 2.4. 16S rRNA Amplicon-Based Sequencing

The bacterial communities in the soil were assessed by sequencing amplicons of the V3–V4 variable region of the 16S rRNA gene, with primer pair 338F (5′-ACT CCT ACG GG AGG CAG CAG-3′) and 806R (5′-GGA CTA CHV GGG TWT CTA AT-3′). The PCR was performed using a TransStart FastPfu DNA polymerase mixture. The reaction mixture (20 μL) was composed of 4 μL of 5x FastPfu buffer, 2 μL of 2.5 mM (each) dNTPs, 0.8 μL of 5 μM Bar-PCR primer F, 0.8 μL of 5 μM primer R, 0.4 μL of FastPfu polymerase, 0.2 μL of BSA, and 10 ng of template DNA. Amplification conditions for the PCR were as follows: 3 min at 98 °C to denature the DNA, followed by 27 cycles of denaturation at 98 °C for 10 s, primer annealing at 60 °C for 30 s, and strand extension at 72 °C for 45 s, followed by 7 min at 72 °C on an ABI GeneAmp 9700 thermocycler (IET, Edison, NJ, USA). Electrophoresis on a 2% agarose gel was used to check the quality of the PCR products and purified using Agencourt AMPure XP beads (Beckman, Brea, CA, USA). The pooled DNA product was used to construct an Illumina paired-end library followed by Illumina-adapter ligation and sequencing by Illumina (MiSeq, PE 2 × 300 bp mode), following the manufacturer's instructions.

## 2.5. Metabarcode Data Processing and Analysis

Paired-end data were demultiplexed into each sample based on the index sequences downloaded from the Illumina MiSeq platform. Hence, the paired-end sequences of each sample were trimmed based on their quality and length using Trimmomatic [18] and FLASH [19] software with the following criteria: (I) reads were trimmed at any site that obtained an average quality score of <20 over a 50 bp sliding window, and the truncated reads shorter than 50 bp were discarded; (II) reads with any mismatch in the barcode, more than two nucleotide mismatches in the primer or containing ambiguous characters were removed; and (III) only sequences that overlapped by more than 10 bp were assembled according to their overlap sequence. Reads that could not be assembled were discarded. The metabarcoding analysis was performed using the online Majorbio Cloud Platform (http://en.majorbio.com/ (accessed on 8 March 2022)). De novo and reference-based chimera detection and removal were performed using Uparse V7.1 (http://drive5.com/uparse/ (accessed on 8 March 2022)). Richness inference and library comparisons were performed using Mothur v.1.9.0 software [20]. Alignments were performed by Mothur using the SILVA bacteria database, with an OTU sequence similarity of 0.97. The taxonomy of each 16S rRNA gene sequence was analyzed by the RDP classifier algorithm (http://rdp.cme.msu.edu/ (accessed on 8 March 2022)) against the Silva 16S rRNA database (Release138

http://www.arb-silva.de (accessed on 8 March 2022)) using a confidence threshold of 70%. The microbiome shared by the different microbial samples was compared and visualized through a Venn diagram plot (R package; https://github.com/vegandevs/vegan (accessed on 5 February 2022)). Circos plots showing microbial structures were performed in Circos 0.67 (http://circos.ca/ (accessed on 8 March 2022)). A polygenetic tree was generated by the FastTree package (V2.1.3, http://www.microbesonline.org/fasttree/ (accessed on 10 March 2022)). PICRUSt (http://huttenhower.sph.harvard.edu/galaxy/ (accessed on 10 March 2022)) and FUNGuild (http://www.funguild.org/ (accessed on 10 March 2022)) were employed to decipher microbial communities and functions. A heatmap plot was constructed to visualize the functional feature abundance profiles using Orange V3.24.1 (https://orange.biolab.si/ (accessed on 11 March 2022)).

## 3. Results

### 3.1. Species Composition Analysis

#### 3.1.1. Taxonomical Representation

The metabarcoding analysis of the 15 soil samples yielded a total of 1,062,960 optimized sequences (i.e., 450,976,362 bases) with an average sequence length of 424 ± 15 bp. A total of 801,936 sequences were valid, and 5490 OTUs were classified and sorted in 46 phylum, 131 classes, 322 orders, 535 families, 1015 genera, and 1935 species quantified by 801,936 total counts across all samples. The average Shannon index for replicates per location was applied to estimate the detected diversity within each sample (i.e., alpha diversity). The index ranged between 0.61 to 6.08, where the highest index was recorded for the rhizosphere area sampled from the agricultural lands, in contrast to the uncultivated area sampled from the reclaimed lands.

Among all samples, the most identified phyla were Actinobacteriota (28%), followed by Proteobacteriota (26%), Firmicutes (14%), Acidobacteriota and Chloroflexi (7%), Gemmatimonadota (5%), Bacteriodota and Crenarchaeota (3%), and Myxococcota (2%), in addition to 37 other phyla with <1% counts (Figure 2C). At the family level, the most represented families were Bacillaceae, Nitrosphaeraceae, Nocardiaceae, Sphingomonadaceae, Burkholderiaceae, Rhizobiaceae, Pseudonocardiacea, Gemmatimonadaceae, and Vicinamibacteriaceae, in addition to 33 other families with counts <1% in at least one of the sampled areas, and 493 families with counts <1% in all samples (Figure 2A,B). Regardless of the land type or the sampled area, the family Bacillaceae was the most represented family among all families, with an almost equal distribution among different categories (land or sample types), followed by the families Nocardioidaceae and Sphingomonadaceae, which were highly present in agricultural and reclaimed lands, specifically more in the rhizosphere than the uncultivated areas. The families Burkholderiaceae and Nitrosphaeraceae were highly present in a desert land and subsequently ranked lower than the other families when the sample area was studied. However, the represented species of this family were present in the uncultivated area more than in the rhizosphere area (Figure 2A,B).

#### 3.1.2. OTUs Inhabiting Rhizosphere Areas

The total OTUs counts was 4033, 3575, and 3327 for agricultural, reclaimed, and desert lands, respectively (Figure 3A). A total of 1984 OTUs co-occurred among the three types of lands regardless of the sampling area, while 779, 1055, and 159 were unique OTUs for agricultural, reclaimed, and desert lands, respectively. The desert lands shared 977 and 243 with agricultural and reclaimed lands separately. All the unique and shared OTUs between desert lands and the other land types were discarded, except for the common shared OTUs that were subject to statistical refining along with the 329 shared OTUs between agricultural and reclaimed lands (Figure 3B).

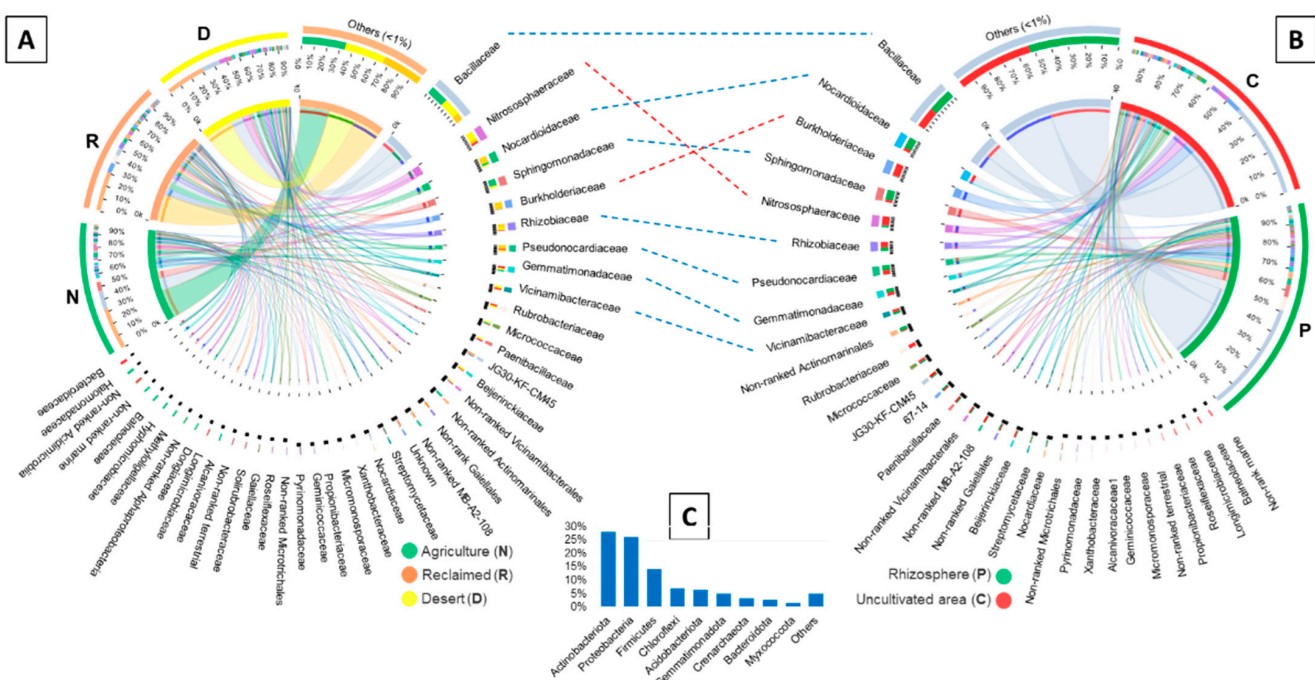

**Figure 2.** Comparative Circos plot showing the two contrasting comparisons ((**A**) land types and (**B**) sample types), while (**C**) is a histogram of the presented phyla in the analyzed samples.

Based on a *t*-test analysis, on one hand, the common shared OTUs among the three land types revealed significant differences in seven microbial phyla, the significant direction was indicated by the positive sign "+" when the phyla were significantly present in cultivated lands (agriculture and/or reclaimed) and vice versa. In details, two phyla (Acidobacteriota and Actinobacterioda) contained two groups each, where the assigned species were significantly present in desertic lands versus cultivated ones. Besides the phylum Myxoccota, the other phyla were significantly present in both cultivated lands, namely, ranked and non-ranked Chloroflexi, nonranked Gemmatimonadota, and Proteobacteria (Figure 3C). On the other hand, the shared OTUs between agricultural and reclaimed lands revealed significant differences in nine phyla and a nonranked bacteria group; the significant direction was indicated by the positive sign "+" when the phyla were significantly present in the rhizosphere area and vice versa. In details, Actinobacteria, Firmicutes, nonranked Acidobacteriota, nonranked Chloroflexi, nonranked Myxococcota, Placentomycetota, and Proteobacteria, in contrast to nonranked Gemmatimonadota and Verrucomicrobiota. By comparing both significant profiles, the negatively significant Actinobacteriota and Acidobacteriota were not present in the rhizosphere, while the nonranked Gemmatimonadota were significantly present in agricultural and reclaimed lands and were absent in the rhizosphere area in contrast to Myxococcota (Figure 3C).

The 74 OTUs unique to the rhizosphere area were quantified below 0.5% of the total sample counts. The assigned OTUs that recorded ≥10 folds were quantified relative to the total of each rhizosphere group (agriculture versus reclaimed). The most represented phyla were Bacteriodota (5.1%:0.3%), Firmicutes (2.4%:0.1%), and Proteobacteria (3.5%:2%) in agricultural and reclaimed lands, respectively. The most represented species with >0.1% in at least one land type, were *Bacteroides coprocola* DSM17136, unclassified Staphylococcus, unclassified Enterococcus, *Brevundimonas vesicularis*, the uncultured bacterium of the family Longimicrobiaceae, an uncultured bacterium of family Muribaculaceae, *Nafulsella turpanensis* ZLM-10, uncultured Epsilonproteobacteria of the order Saccharimonadales, *Rhodococcus rhodochrous*, nonranked Saccharimonadales, and uncultured bacterium of family Cellvibrionaceae.

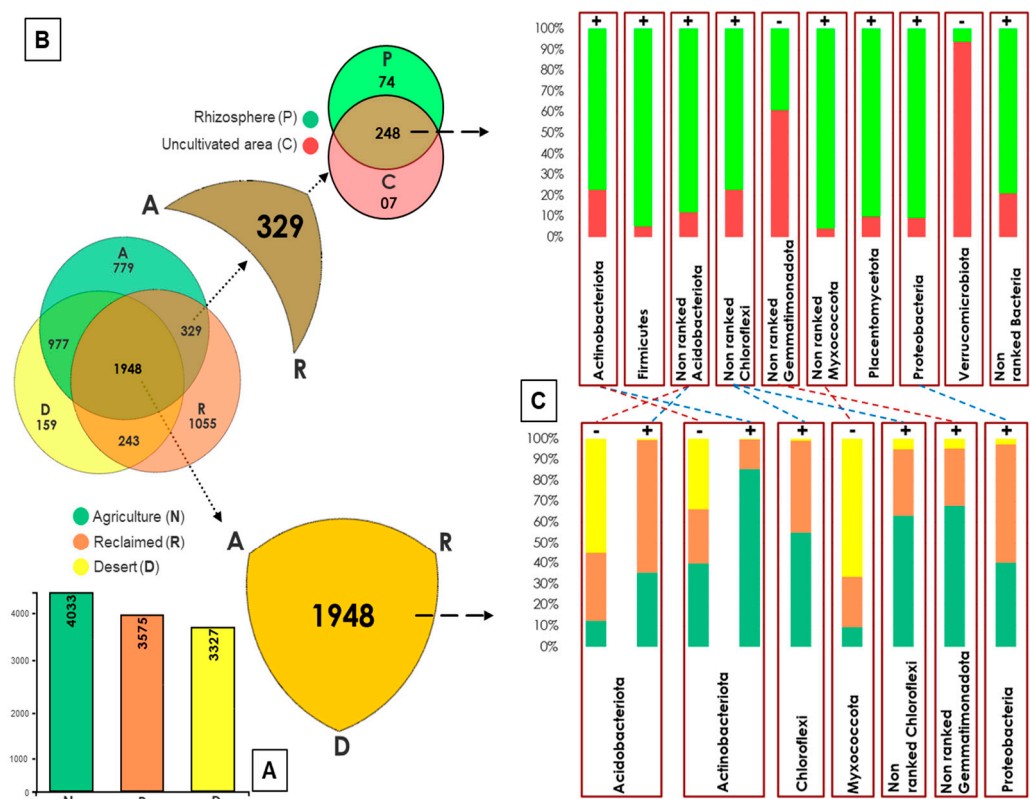

**Figure 3.** A comparative Venn diagram between three land types (agriculture "N", reclaimed "R", and desert "D") based on species counts (**A**) and the total count for each land type is shown as a histogram (**B**). The common OTUs were compared statistically and shown in (**C**). The significance direction is indicated by the positive sign "+" when the phyla are significantly present in cultivated lands (agricultural and/or reclaimed; rhizosphere area) and vice versa.

### 3.2. Prediction Analysis

#### 3.2.1. Phenotypic Prediction

Based on the recorded metadata for microbial species in databases, seven categories were defined. The most represented categories among all samples were biofilm-forming, Gram-negative, and aerobic bacteria, respectively. The phenotypic profiles of the rhizosphere and uncultivated areas were compared and controlled by the land types. The aerobic and biofilm-forming bacteria were found to be highly present in the rhizosphere and were more present in the cultivated lands (agricultural and reclaimed) than the desert land. While the bacteria containing mobile elements, potentially pathogenic or tolerant to stress, were highly present in the rhizosphere, more so than in the uncultivated area, they were only highly present in the reclaimed lands rather than the agricultural or desert lands (Figure 4A).

When the species-phenotype contribution was revised at the genus level, the species belonging to order Actinomarinales (nonranked species), and the genus Sphingomonas were highly contributing to both phenotypes (aerobic and biofilm-forming bacteria) in the rhizosphere in contrast to the uncultivated area. Equally, the species belonging to the genus Nocardioides were highly contributing as aerobic bacteria only. Species belonging to genera Ralostonia and Rubrobacter were the most contributing aerobic and biofilm-forming bacteria in the uncultivated areas and were absent in the rhizosphere areas. Additionally, species belonging to the family Vicinamibacteraceae (nonranked) were one of the most contributing biofilm-forming bacteria in both rhizosphere and uncultivated areas (Figure 4B).

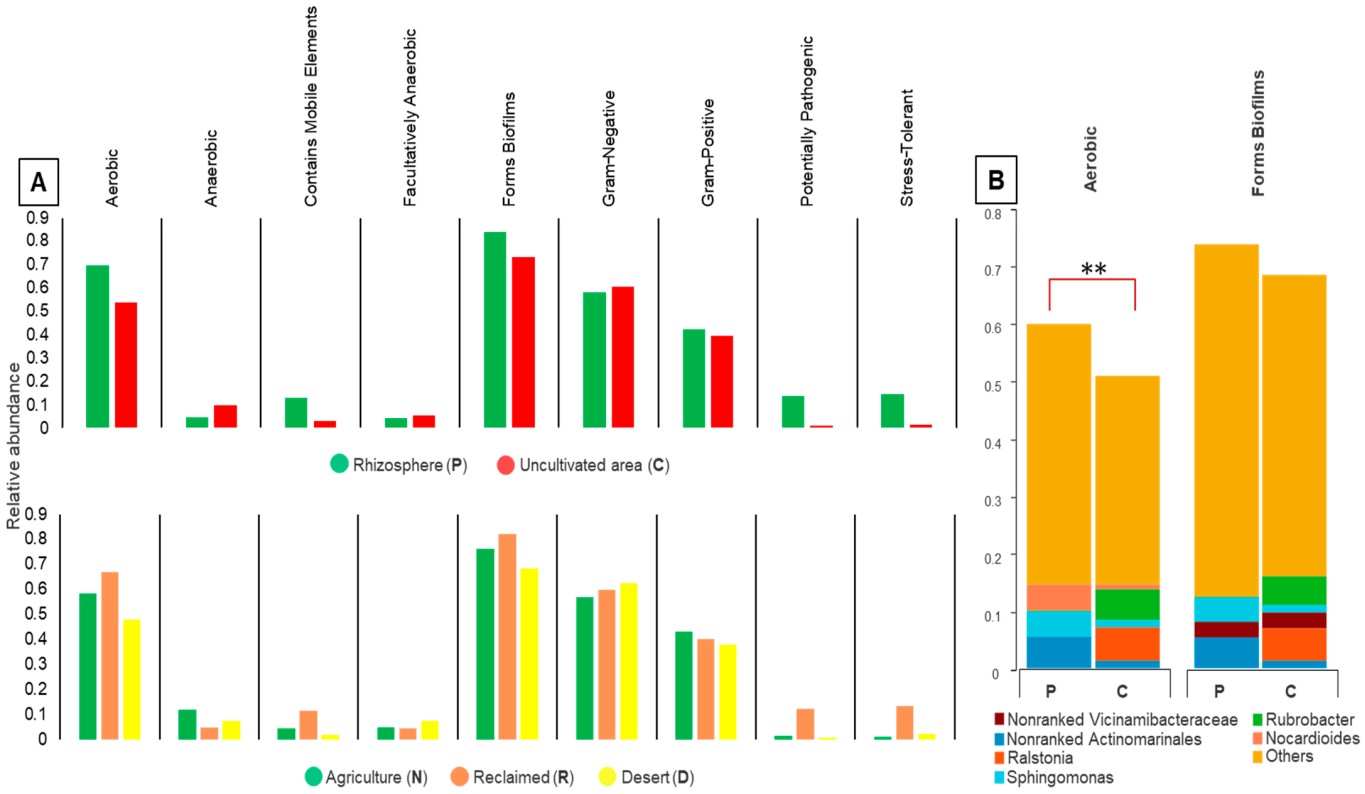

**Figure 4.** Histogram of the predicted phenotypes for the microbial community detected among all samples (land and sample types, (**A**)); the species representing the significant phenotypic profiles are presented comparing the rhizosphere to the uncultivated area (**B**). ** *p*-value > 0.001.

### 3.2.2. Functional Prediction

Among all samples, the most present cluster of ortholog genes (COG) was the amino acid transportation and metabolism, counting at an average abundance of 30 M genes, followed by the cluster of genes with unknown function, the genes of general function prediction. The energy production and conversion genes in addition to the transcription genes were among the highly abundant clusters (Figure 5A). Particularly, the gene cluster described as dehydrogenase reductase (COG1028), transcriptions regulators (COG1309), and major facilitator (COG2814) were the most abundant genes in the rhizosphere area samples compared to the uncultivated samples. With less abundancy, the histidine kinase (COG0642), alpha-beta hydrolase (COG0596), RNA polymerase (COG1595), acyl-CoA dehydrogenase (COG1960), transcriptional regulator (COG0583), glycosyl transferase (COG0438), the transcriptional regulator "LuxR family" (COG2197) and methyltransferase required for the conversion of demethylmenaquinone (DMKH2) to menaquinone (MKH2) (COG2226), were detected.

By comparing the gene enrichment results between desert land versus the cultivated lands (agricultural and reclaimed), the most enriched genes were forming part of the citrate cycle (TCA cycle, Krebs cycle; M00009), followed by the reductive citrate cycle (Arnon–Buchanan cycle; M00173), NADH: quinone oxidoreductase (M00144), glycolysis (Embden–Meyerhof pathway; M00001), citrate cycle, second carbon oxidation (M00011), phenylacetate degradation (M00878) dicarboxylate-hydroxybutyrate cycle (M00374), and beta-oxidation (M00087; Figure 5B).

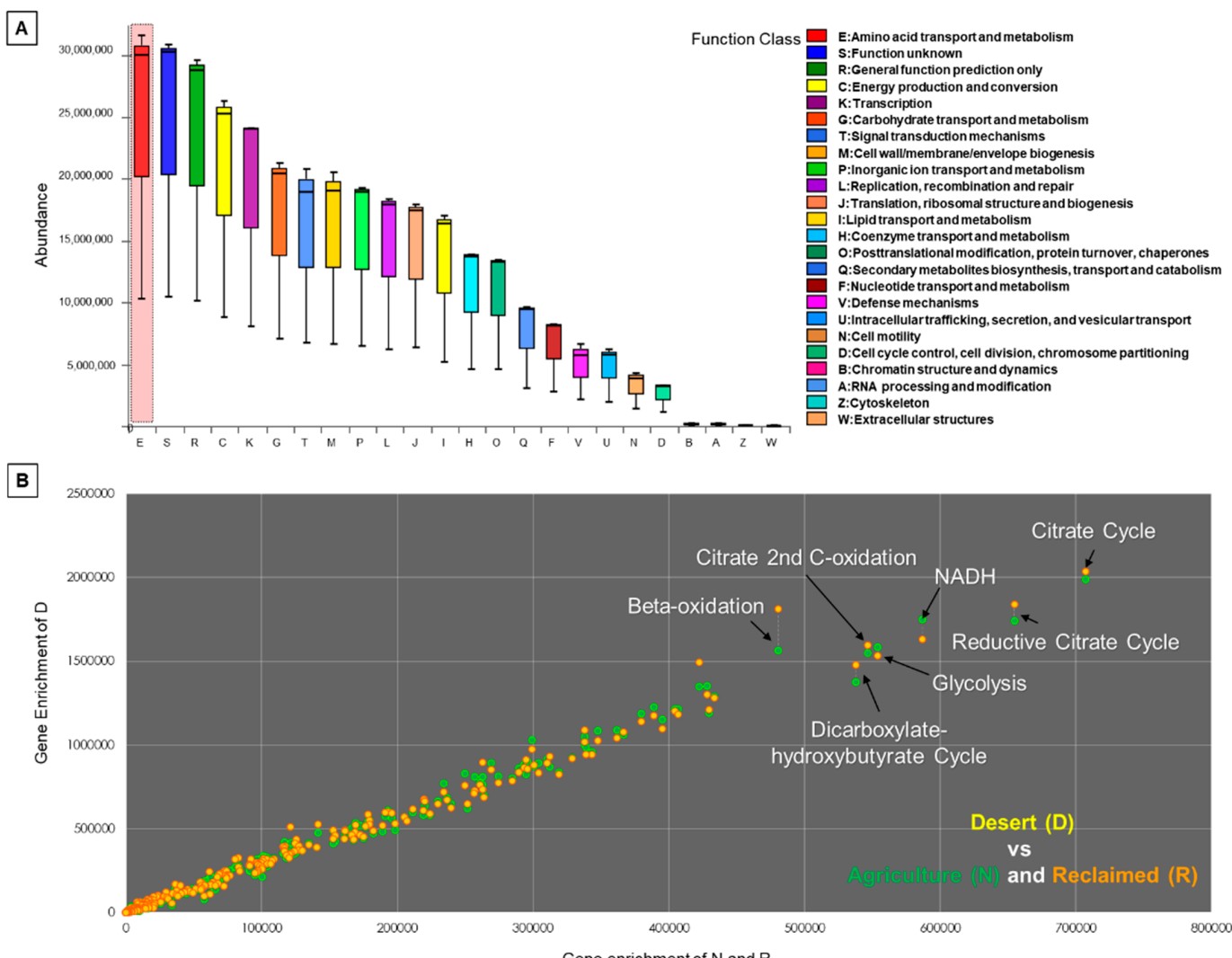

**Figure 5.** The functional prediction charts include the clusters of orthologue genes (COGs) for all samples (**A**) and the gene enrichment dot plot of the cultivated lands versus the desert land (**B**).

### 3.3. Association Analysis

#### 3.3.1. Co-Occurrence Correlation Network

The correlation-based distance matrix revealed several correlation blocks of positively and negatively correlated bacterial genera. The blocks were defined by a descending letter for the block size. The D block was represented by species belonging to the order Actinomarinales (nonranked species) correlated to species belonging to genera Chelativorans and Halomonas, as well as nonranked species of the PAUC43f marine group; this block along with species of the genera Rhodococcus and Ralstonia were negatively correlated to blocks A, B, and C, and marked and labeled as E and F. The A correlation block represented the highest group of correlated species; for example, species belonging to the genera Sphingomonas, Rubrobacter, and species belonging to the family Vicinamibacteraceae (nonranked) and family Gemmatimonadaceae (nonranked). The B block included species of the families Nitrososphaeraceae (nonranked), Roseiflexacea (nonranked), along with species belonging to the genera Nitrospira, Aeromicrobium, Dongia, Nocardioides, and Pseudonocardia. The C correlation block was formed by species belonging to the genera Ammoniphilus, Streptomyces, Arthrobacter, Paenibacillus, and Bacillus (Figure 6).

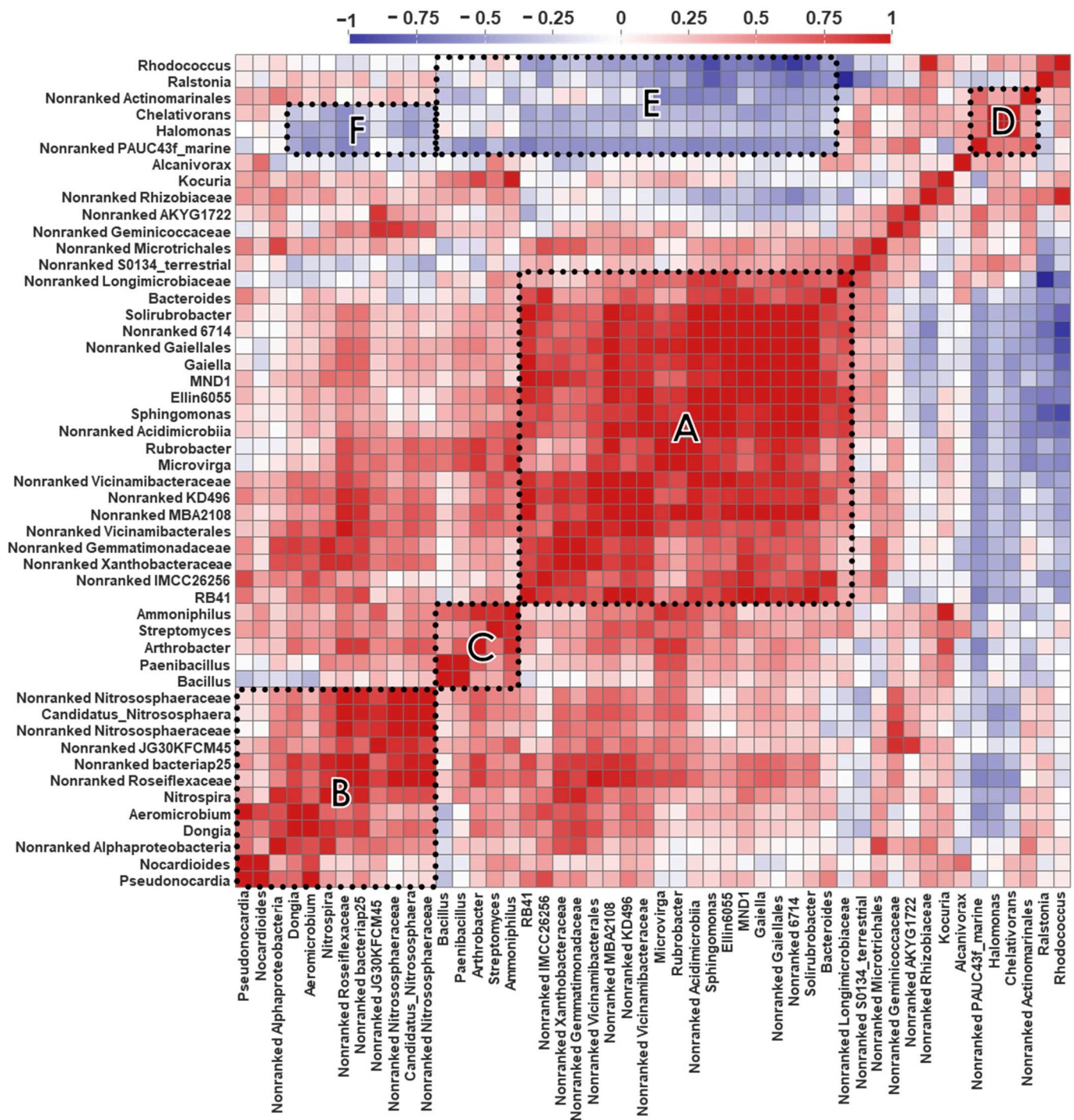

**Figure 6.** Heatmap correlation distance matrix among all the recorded species. Correlation blocks are defined by letters: A–D represent the positively correlated blocks (0 > r-value ≥ 1) while E and F represent the negatively correlated blocks (−1 ≥ r-value > 0).

### 3.3.2. Association to Soil Properties

Major elements were detected from the sampling area, including P, Fe, K, Mn, Zn, and Cu, in addition to the soil organic matter (OM). K was the highest major component among all with an average of 277.21 ± 76.82. The following major components of the soil were Fe and Mn with an average of 9.54 ± 2.75 and 9.28 ± 4.08, respectively. The averages of P, Zn, and Cu were 5.18 ± 4.67, 3.59 ± 1.29, and 4.34 ± 71, respectively, while the lowest average was found for the soil organic matter 0.06 ± 0.04 among all land types. The reclaimed

land soil profile contained the highest values in P (10.55), Fe (12.71), Zn (5.08), Cu (4.95), moderate values in Mn (8.07), and lowest values in K (194.31). The agricultural land was approximately equal to the desert land in P (~2.5), Fe (~8), Zn (~3), and Cu (~4). The agricultural land was only the highest in OM (0.09), equal to that of the reclaimed land (0.08). The desert lands were, remarkably, the highest in K (345.99) and Mn (13.84). The Mn and Fe profiles were inversely proportional in the desert (i.e., Fe was lower in the desert than Mn) compared to cultivated lands regardless of the quantity (i.e., Fe was higher in agricultural and reclaimed lands than Mn; Figure 7A).

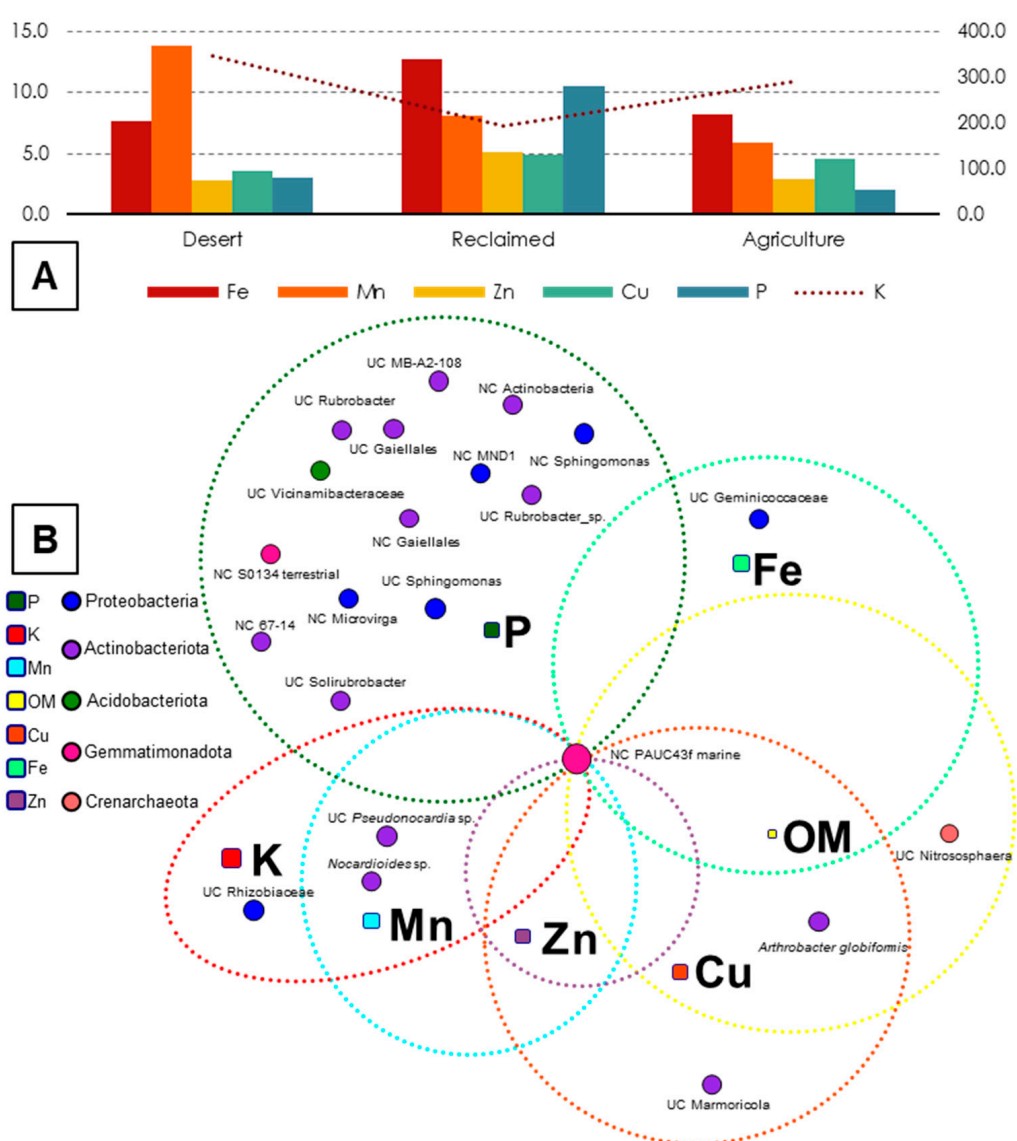

**Figure 7.** The metal composition measured per land type (**A**) and the association of species to each metal (**B**).

The species' statistical association to soil elements was analyzed, where P was the most influential element to the number of associated species (14 species), followed by Mn and K associated with unknown and uncultured species of the genus Nocadriodes and Psuedonocardia, respectively. *Arthrobacter globiformis* was associated with both Cu and OM values. Additionally, uncultured archaea of the genus Nitrososphaera and uncultured bacteria of the genus Marmoricola were associated with OM and Cu, respectively. K was associated with uncultured bacteria of the family Rhizobiaceae and Fe with uncultured

bacteria of the family Geminicoccaceae. Zn was not associated with any of the detected microorganisms (Figure 7B).

## 4. Discussion

The current study aimed to identify the rhizosphere-associated bacteria by following a specific sampling design. The multi-plant sampling from different land types was the main feature of our analysis, using the desert as a general control to omit and discard desert-related species from our analysis. Additionally, the rhizosphere areas in the cultivated lands (whether the usual agriculture or newly reclaimed lands) were controlled by the uncultivated areas within each land. The application of different statistical approaches was meant to resolve as many relevant species as possible associated with the plant rhizosphere. The general observation was the species taxonomical status of the relevant species, in which the majority were unidentified, unknown, unranked, and/or uncultivated bacteria. Even though the lack of this knowledge would fog our survey, it also opened an interesting window for the traditional isolation of the relevant species for future studies. Thus, the interpretations at the genus level are the dominating aspect of this discussion.

Carbon fixation is considered one of the main biochemical processes in the biosphere, providing the carbon-constructing blocks for entirely living organisms [21]. The species belonging to the order Vicinamibacterales were recently recorded as potential carbon fixation bacteria [22]. Nitrification consists of the biological oxidation of ammonia ($NH_3$) to nitrite ($NO_2^-$), which is carried out by NH3-oxidizing archaea (AOA) and bacteria (AOB), combined with the oxidation of $NO_2^-$ to nitrate ($NO_3^-$ is a form that can be used by plants) are carried out by phylogenetically diverse $NO_2^-$-oxidizing bacteria (NOB) [23]. Species belonging to the phylum Chloroflexi, order Rhizobiales, the family Nitrosomonadaceae, and genus Nocardioides are known NOBs that gain energy from the oxidation of nitrite to nitrate, and some are capable to convert the atmospheric $N_2$ to $NH_3$ [24]. Actinomarinales are well-known aquatic-system-associated Actinobacteria, particularly in nutrient-limited locations that require high surface-to-volume ratios, with a potential capability to reduce carbonate and nitrate [25]. Cyclobacteriaceae is considered a member of the class "Sphingobacteriia" that belongs to the phylum Bacteroidetes. The members of the family were reported from diverse inland and off-land habitats [26]. Under aerobic conditions, they can oxidize carbohydrates (one species ferments glucose) but do not produce indole, while some taxa reduce nitrate to nitrite [27]. The phylum Planctomycetota is highly abundant in sediments from diverse geographical environments, they are reported to be the dominant group in oxygen minimum zones (OMZ). The members of one of this phylum family, namely, Phycisphaeraceae, can oxidize ammonium under oxygen minimum conditions and are collectively named anammox [28]. Iron is a prevalent redox-active metal element present on the Earth and occurs in two oxidation states ($Fe_2^+$ and $Fe_3^+$) in nature. Some microorganisms drive the oxidation of $Fe_2^+$ to gain energy for growth, with molecular oxygen or $NO_3^-$ such as the electron acceptor [29]. Acidimicrobiia is a deep-rooting lineage that belongs to the phylum Actinobacteria, which can oxidize ferrous iron at comparatively fast rates and is responsible for the regeneration of ferric iron in the acidic ecosystem [30]. On the contrary, Bryobacter includes chemoheterotrophs, aerobes, and facultative anaerobes and is capable of reducing Fe (III). Members that belong to these genera have usually been isolated from acidic wetlands but grow better through mildly acidic conditions [31]. In the current analysis, despite the presence of alternative electron transports in the soil (e.g., Fe and Mn) and bacterial species with metal oxidation and reduction capabilities that may promote the anaerobic respiration [32], the most enriched functional orthologue cluster of genes were related to the aerobic respiratory bacterial phenotypic profile, a phenotype that was significantly present in the rhizosphere areas when compared to uncultivated areas.

Many of the surveyed species belonging to the rhizosphere area were previously reported as decomposers of soil organic matter, producers of secondary metabolites, soil remediators, and plant-growth-promoting bacteria. Plant-promoting bacteria can act on the regulation of plant metabolism by producing or stimulating the production of various

phytohormones that enhance intensive growth for seedlings such as auxins indole-3-byturic Acid (IBA), (indole-3-acetic acid (IAA), and phenyl acetic acid (PAA) [33].

Species belonging to the phylum Actinomycetota were the most represented in the plant rhizosphere, especially the class Actinobacteria. This class represents an enormous group of microorganisms that can produce a varied range of secondary metabolites, involving surfactants. They also contribute to the rotation of soil components into organic components through the decomposition of a complex combination of organic matter in lifeless plants and animals, in addition to fungal material [34]. *Rhodococcus rhodochrous* species that belongs to the family Nocardiaceae is well-known to cometabolize difficult-to-degrade hydrocarbons [35]. Members of the family Gaiellaceae are naturally producing many different antibiotics and contribute to global carbon cycling through the decomposition of soil organic matter, increase plant productivity, and are widely known as prolific producers of bioactive compounds essential for humans and animal health [36]. The genus Streptomyces, which is known as the most abundant and certainly the most important Actinomycetes, is considered a good source of antibiotics, bioactive compounds, and extracellular enzymes. It plays a major role in nutrient cycling and, more significantly, because of the general propensity of members of the genus to produce secondary metabolites of biotechnological and clinical importance. The importance of Streptomyces stems from its biocontrol, plant-growth-promoting, and being efficient as a biofertilizer [37]. Members of the genus Actinoplanes which belong to the family Micromonosporaceae are prolific sources of novel enzymes, antibiotics, and other bioactive compounds [38]. The genus Micromonospora within the same family has a great potential for producing secondary metabolites. Micromonospora species function in biocontrol, plant growth promotion, root ecology, and the breakdown of plant cell wall material [39]. Functions that predicted the order Microtrichales were related to gluconeogenesis and/or glycolysis, chlorophyll and porphyrin metabolism, transcription factors, and photosynthetic proteins [40], while the relative abundances of the Solirubrobacteraceae family were found positively correlated with cultivated plant growth [41]. Within the family Nocardioidaceae, the genus Nocardioides is an aerobic, motile, or nonmotile genus that plays an important role in the degradation of di-2-ethylhexyl phthalate (DEHP) in natural soil environments [42], and the decomposition of various pollutants such as alkanes, pyridine, phenols, phenanthrene, etc. [43]. Many organisms within the genus Nocardioides show biodegradative activities, exhibiting the capacity to metabolize recalcitrant and toxic environmental pollutants, in addition to secreting a range of extracellular enzymes [44]. The genus Aeromicrobium produces a wide diversity of secondary metabolites as major compounds with antibacterial activity, and they could be potential indicators for disease repression [45]. Pseudonocardia in the family Pseudonocardiaceae are well-known to degrade 1,4-dioxane as the sole carbon and energy source in addition to degrading tetrahydrofuran (THF). It has an important role in biotechnology due to the production of secondary metabolites, some of which have antibacterial and antifungal effects and help in the decomposition of the organic matter of dead organisms [46].

Within the phylum, Pseudomonadota (synonym Proteobacteria), the second most represented phylum, genus Roseomonas, is a genus of aerobic, motile bacteria of the family Acetobacteraceae, known as acetic acid bacteria that can produce specific secondary metabolites, i.e., gentamycin and asukamycic [47]. The species that belong to the Sphingomonas genus have many functions ranging from remediation of environmental contaminations to producing highly beneficial phytohormones, for example, sphingan and gellan gum. The degradation of organometallic compounds improves plant growth during stress conditions such as salinity, drought, and heavy metals in agricultural soil by producing plant growth hormones, e.g., gibberellins and indole acetic acid [48]. Bacteria that belong to the genus Novosphingobium regularly participate in the biodegradation of aromatic compounds such as aniline, phenol, 4-chlorobenzene, nitrobenzene, and pyrene, phenanthrene, dibenzofuran, carbofuran, and estrogen [49]. Cellvibrionaceae have a terrestrial origin, related to soil and decaying plant materials; however, Cellvibrionaceae are marine bacteria and

display a slightly halophilic behavior. Most species in this family possess a large variety of polysaccharide-degrading abilities and their genomes contain dozens of CAZyme (carbohydrate-active enzyme) genes, enabling the hydrolysis of cellulose, agar, carrageenan, xylan, starch, chitin, and several other polysaccharides [50].

Additional phyla were detected as part of the rhizosphere microbial community. In the phylum Acidobacteriota, the bacterial species have genes that probably help survival as well as competitive colonization in the rhizosphere, leading to the establishment of beneficial relationships with plants, the regulation of biogeochemical cycles, the decomposition of biopolymers, exopolysaccharide secretion, and plant growth promotion [51]. Blastocatella is important in pharmaceutical wastewater treatment plants, and it contributes to ammonium nitrogen removal [52]. Members that belong to the family Anaerolineaceae within the phylum Chloroflexi are anaerobic microbes, that can coexist with methane metabolism microbes and are important organic matter degraders under anoxic conditions, while methane metabolism is used for the bioremediation of soil contaminated with Cd and promotes the precipitation of soluble Cd [53]. The Bacillales of the phylum Firmicutes enhance plant growth through the production of ACC deaminase and pathogen suppression [54]. Within the phylum Myxococcota, the members of the family Myxococcaceae are broadly distributed in soil and also exist in freshwater in addition to the marine environment, with the ability to produce diverse secondary metabolites acting as antimicrobials, antiparasitics, antivirals, cytotoxins, and antiblood coagulants [55]. Members of the class Polyangia are well-known for their extraordinary social lifestyle and diverse novel gene clusters of secondary metabolites in soil [56]. Saccharimonadales that belongs to the Patescibacteria phylum have small genomes with supposed parasitic or symbiotic lifestyle. Saccharimonadales are known as candidate bioindicators of high P availability and considered the dominant bacteria in salt stress or organic enriched sludge which might degrade plastics and also show synergistic effects through the nitrogen cycling-related genes. Saccharimonadales are fast-growers and use sugars for energy metabolism [57]. Within the Verrucomicrobiota phylum, the members of the Pedosphaeraceae family were found to tolerate Cd biotoxicity and are used for the optimization of phytoremediation in Cd-contaminated sediment [58].

Gemmatimonadota is known as the eighth-most abundant bacterial phylum in soils, representing about 1–2% of the soil bacteria worldwide. They are typically short rods and are rich in soils, wherever they seem to be frequently associated with the plants and the rhizosphere as well as freshwaters, wastewater treatment plants, biofilms, and sediments and are capable of anoxygenic photosynthesis [59]. Longimicrobiaceae are found in Mediterranean forest soil and is a member of the order Longimicrobiales within the class Longimicrobia. They are considered nonmotile, Gram-negative, short-to-long rod-shaped bacteria with anaerobic chemoorganoheterotrophic metabolism. Members of the family Longimicrobiaceae as well as the members of the family Cyclobacteriaceae (the phylum Bacteroidota) facilitate phosphate solubilization in the soil [60]. Some of these bacterial taxa were also frequently found in other semiarid soils with low organic matter content and adapted to extreme conditions, such as Blastococcus or uncultured members of the family Longimicrobiaceae and the genus Staphylococcus, which appeared exclusively in no-organic matter control soils [60].

## 5. Conclusions

Based on the function prediction analysis, it appears that carbon-fixation- and nitrification-capable bacteria were abundant in the rhizosphere area. It included previously reported aerobic bacteria, decomposers of soil organic matter, soil remediators, and producers of a varied range of secondary metabolites acting as antimicrobials, antiparasitic, antivirals, cytotoxins, and antiblood coagulants. Moreover, it also included bacteria that are capable of degrading organometallic compounds as well as improving plant growth during stress conditions such as salinity, drought, and heavy metals in agricultural soils, by producing plant growth hormones. Additional bacteria were highlighted for their

biodegradative activities and bioindicators of high P availability. At the functional level, the genes enriching gluconeogenesis and/or glycolysis pathway, chlorophyll and porphyrin metabolism, transcription factors, and photosynthetic proteins characterized the microbial community of the rhizosphere area, as well as genes that probably help in the survival and competitive rhizosphere colonization. A further RNAseq analysis completed the functional overview of the studied microbial community, validated the predicted function, and provided insights into the microbial interactions within the rhizosphere area. Even though the surveyed microbial list included several unknown species, it opens a prospect for in vitro isolation and cultivation of new species, to identify and characterize bacteria capable of establishing beneficial relationships with plants (i.e., promoting growth) and provide potential candidates as biofertilizers.

**Author Contributions:** Conceptualization, S.M.R., M.M. and M.A.-S.R.; methodology, M.M.; software, M.M. and Y.G.F.; validation, M.H.K.; formal analysis, M.M.; investigation, M.H.K.; resources, M.H.K.; data curation, M.M.; writing—original draft preparation, M.M., Y.G.F. and M.H.K.; writing—review and editing, M.M., A.H.A. and M.A.-S.R.; visualization, M.M.; supervision, S.M.R., A.H.A. and M.A.-S.R.; project administration, M.M.; funding acquisition, M.H.K. All authors have read and agreed to the published version of the manuscript.

**Funding:** This research received no external funding.

**Institutional Review Board Statement:** Not applicable.

**Informed Consent Statement:** Not applicable.

**Data Availability Statement:** Metabarcoding raw data were deposited to the National Center for Biotechnology Information (NCBI) under the Bio-Project number PRJNA850534 (https://www.ncbi.nlm.nih.gov/bioproject/PRJNA850534/).

**Acknowledgments:** The authors are grateful for the support of Mostafa Nafaa in the sampling and transportation of the material used in the current study.

**Conflicts of Interest:** The authors declare no conflict of interest.

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
