# Peer review of "Rhizosphere-Associated Microbiome Profile of Agricultural Reclaimed Lands in Egypt"

_agronomy, doi:10.3390/agronomy12102543_

Round 1

Reviewer 1 Report

The manuscript "Rhizosphere-Associated Microbiome Profile of Agriculture Re-2 claimed Lands in Egypt”, aims to identify the bacterial profile in agriculture soil samples and relate rhizosphere in newly reclaimed land and habitual cultivated ones.

The manuscript falls well within the scope of the Agronomy journal. The experiment was carried out in 2021 considering three areas: agricultural lands, horticulture, and desert land management. The bacterial profile was assayed by the metabarcoding technique. 

The topic is interesting and the results of this study might be useful to improve sustainable soil management in this region. The study has well been designed and conducted, but it needs to be better described and discussed. Some parts could be improved. So, I suggest Major revisions

Introduction: 

There is not a clear hypothesis in this study. Please add.  

Material and methods section: 

Introduction 

Lines 85-88: the aims of the study, described in these lines and also on lines 9-11 of the abstract, did not consider desert land. This was in contrast with those described in the discussion section, see lines 337-342. Please check.

Line 93: According to tab. 1, for agricultural soils and reclaimed land different land uses, were considered (apricot, citrus..etc). I suggest adding after this line, an explanatory sentence: for example “ for agricultural soils and reclaimed land, three subplots covered by different plantation types were considered (Tab.1) “This allows to help the reader. 

Line 97: Add “(Figure 1)” after “some type”.

Lines 97-100: the authors used the terms “plots”, “sites, and “location” in the same sentences. I understand that you have 3 locations, in each one of which you choose some (how many?) plots where soil and plant samples were taken. Is it correct? Moreover, the author first described that soils from some sites were mixed, and then in the following line that three replicates were obtained. In the actual form, this description may generate confusion in the readers, so it is necessary to clarify. 

Line 109: stored at room temperature? For DNA analysis? Are sure? Moreover, this sentence is in contrast with the following in line 112. Please check.

Line 113:About texture: the author assay texture but did not show this data in the result and discussion section. Please check.

Lines 167-175: Are data not shown? Please clarify

Line 179: Add (figure 2) after “with <1% counts”.

Line 309-323: Is it possible to show soil physicochemical properties in a table, also adding in the supplementary material. This could help to follow better the results description of this part. 

Moreover, on lines 310-311, for the first time, the authors have shown data about soil nutrients (P, Fe, K, Mn… etc). Are those literature data? Are obtained in this study? In the first case, a citation is necessary, in the latter case this method is not shown in the material and methods section. 

 Discussion: see comment for lines 85-88. Moreover, this part should be carefully revised to improve clarity.

Conclusion section: 

This part is too long and seems a list of bacterial functions, rather than a synthesis of the study data. However, a reader should find immediately the main results of the study, in the conclusions

 Tables 

In my opinion, tables and figures caption should be self-understanding, but at the same time concise. 

This is not true for the Table 1 caption. I suggest verifying this form: ”Lands type, plantation types and sample types (rhizosphere and uncultivated areas) and sample code”. Moreover, in this caption, the three lands assayed were described by the terms “land types”, while in the text as “location”. See comment regarding lines 97-100.

 Figure 1: “ land types” ? see the previous comment

Figure 2 and Figure 3: I suggest adding the letters to each panel of this figure and then citing them in the text to allow the reader to follow the presentation of the results section better.  

Author Response

Dear Editor,

Thank you very much for your time and effort handling our manuscript. Please find the responses to reviewer 1, as follows:

Reviewer 1:

The manuscript "Rhizosphere-Associated Microbiome Profile of Agriculture Reclaimed Lands in Egypt”, aims to identify the bacterial profile in agriculture soil samples and relate rhizosphere in newly reclaimed land and habitual cultivated ones. The manuscript falls well within the scope of the Agronomy journal. The experiment was carried out in 2021 considering three areas: agricultural lands, horticulture, and desert land management. The bacterial profile was assayed by the metabarcoding technique. The topic is interesting, and the results of this study might be useful to improve sustainable soil management in this region. The study has well been designed and conducted, but it needs to be better described and discussed. Some parts could be improved. So, I suggest Major revisions

Response: Thank you very much for your effort and time, we welcome all the proposed comments and suggestions. Please find our point-to-point responses as follows:

Introduction:

There is not a clear hypothesis in this study. Please add.

Response: Thanks for your comment

The following part was rephrased and modified in the revised version (L85-91).

The current study aimed to profile the bacterial community of agriculture-related soil samples using the metabarcoding technique. The study was designed to explore and compare the relevant rhizosphere bacteria associated with plant cultivations by contrasting three land types (newly reclaimed lands and habitual cultivated ones, controlled by uncultivated desert lands). The complete metabarcoding profiling would test whether the bacteria inhabiting the rhizosphere area of cultivated plants would be associated with the agriculture practice regardless of the soil type or the microbial source. And eventually recommend the identified beneficial bacteria as biofertilizers in any land type.

Material and methods section:

Introduction

Lines 85-88: the aims of the study, described in these lines and also on lines 9-11 of the abstract, did not consider desert land. This was in contrast with those described in the discussion section, see lines 337-342. Please check.

Response: Thank you for the comment, the desert land was used as a control to discard and omit irrelevant species and to help in identifying more accurately the species that inhabit both cultivated land and newly reclaimed land. We believed that the abstract nature wouldn’t require such information in contrast to the discussion part which required an explanation for the sampling design and purpose.

Line 93: According to tab. 1, for agricultural soils and reclaimed land different land uses, were considered (apricot, citrus… etc). I suggest adding after this line, an explanatory sentence: for example, “for agricultural soils and reclaimed land, three subplots covered by different plantation types were considered (Tab.1) “This allows to help the reader.

Response: Thanks for your suggestion. The suggested editing was noted and changed accordingly

Line 97: Add “(Figure 1)” after “some type”.

Response: Noted and changed accordingly

Lines 97-100: the authors used the terms “plots”, “sites, and “location” in the same sentences. I understand that you have 3 locations, in each one of which you choose some (how many?) plots where soil and plant samples were taken. Is it correct? Moreover, the author first described those soils from some sites were mixed, and then in the following line those three replicates were obtained. In the actual form, this description may generate confusion in the readers, so it is necessary to clarify.

Response: Sorry for the confusion and thank you for pointing this issue. We carefully rephrased this part for clarity in the revised version of the manuscript.

Line 109: stored at room temperature? For DNA analysis? Are sure? Moreover, this sentence is in contrast with the following in line 112. Please check.

Response: In line 109, the samples were stored at room temperature immediately after collection and until examination and to be separated and aliquoted for further analysis. In line 112, the separation of material was already done, and the part that was used for DNA extraction was frozen in order to preserve the microbial content form changing under laboratory conditions until the extraction process starting.

Line 113: About texture: the author assay texture but did not show this data in the result and discussion section. Please check.

Response: We are sorry for this mistake. We had some incomplete analysis and didn’t correct the text. The text was rewritten carefully stating the finally used and correct analytical methods.

Lines 167-175: Are data not shown? Please clarify

Response: This part is refered to the metabarcoding sequencing, and the sentence was rephrased accordingly. We mentioned the sequencing stats acquired from the analyzed samples, and the diversity analysis showed the species ranking in numbers, which was mentioned in the text.

Line 179: Add (figure 2) after “with <1% counts”.

Response: We believe that “Figure 1” is the intentioned figure to be added to the sentence, thus the requested change was Noted and added accordingly.

Line 309-323: Is it possible to show soil physicochemical properties in a table, also adding in the supplementary material. This could help to follow better the results description of this part.

Response: Thank you for your comment, we believe that the numbers are presented as averages in form of histogram (Figure 6A) comparing all the studied lands covering the same information as the table would give, thus the table idea was avoided from the beinging to avoid data repeatation. If it is still not clear, we can add in the next round of revisions.

Moreover, on lines 310-311, for the first time, the authors have shown data about soil nutrients (P, Fe, K, Mn… etc). Are those literature data? Are obtained in this study? In the first case, a citation is necessary, in the latter case this method is not shown in the material and methods section.

Response: Thanks for your comment, the data was measured by us, not a litrature data, thus the citation will not be required, and the M&M part was modified as previously mentioned.

Discussion: see comment for lines 85-88. Moreover, this part should be carefully revised to improve clarity.

Response: Thanks for your comment. We revised the text to make sure the point of disscusion was clear, which is related to the justification of the study design and the reason why we used desert soil and uncultivated areas as control combined with different statistical methods to resolve as much as possible of rhizosphere-related species; in addition to highlighting that the majority of the detected species were unknown or unidentified opeining a window for prespective studies.

Conclusion section:

This part is too long and seems a list of bacterial functions, rather than a synthesis of the study data. However, a reader should find immediately the main results of the study, in the conclusions

Response: Thank you very much for the comment, we have shorten the conclusion considering the raised comment, however, part of the results was revealed by the dissused litrature that highlight the characteritics and the profile of the detected microbial community in the rhizosphere, thus is form part of the main results of the study and we have included only the key findings-by-litreature in the conclusion part.

Tables

In my opinion, tables and figures caption should be self-understanding, but at the same time concise.

This is not true for the Table 1 caption. I suggest verifying this form: ”Lands type, plantation types and sample types (rhizosphere and uncultivated areas) and sample code”. Moreover, in this caption, the three lands assayed were described by the terms “land types”, while in the text as “location”. See comment regarding lines 97-100.

Response: Thanks for your comment. Noted and changed accordingly.

Figure 1: “land types” ? see the previous comment

Response: Considering all the comments related to this points, land types should be kept intact to be homogenious with the othe parts.

Figure 2 and Figure 3: I suggest adding the letters to each panel of this figure and then citing them in the text to allow the reader to follow the presentation of the results section better.

Response: The suggestion is most welcomed, and it was applied to figures 2 and 3 and cited accordingly in the text.

We are very gratful to all the effort and time dedicated to our maunscript,

Best regards

M Magdy

Reviewer 2 Report

The manuscript agronomy-1717855 evaluates the rhizosphere-associated microbiome profile of agricultural lands located in three Egyptian major areas. The aim of the study was to identify sources of microorganisms to be used as biofertilizers. The authors studied microbial communities by 16S rDNA metabarcoding and applied bioinformatics and statistics to investigate samples and predict metagenome functions. The study is interesting and could add knowledge to the field. I would improve only minor aspects. See specific comments below.

  • Keywords: they should not match the words present in the title.
  • Introduction: The introduction correctly places the study in context. I would improve the clarity of the purpose of the study and the working hypotheses being tested:

L86: Clarify the purpose of the "relevant bacteria" you searched for.

L88: Add working hypotheses.

  • Materials and Methods: The authors described in sufficient detail the methods used. However, I would briefly describe the methods and instruments used for the laboratory methods stated in line 114. 
  • Results: The results description is clear. Some sentences need a moderate English revision.
  • Discussion: The authors correctly discussed the results from the perspective of previous studies in the broadest context possible.
  • Conclusions: The section is appropriate as the manuscript contains many elements. However, since the study was explorative, the authors should mention how they will continue future studies based on the findings obtained. I would also underline that the functional analysis carried out is only predictive. Some statements on the importance of an RNA-sequencing analysis to study the activity of the rhizosphere microbiome must be provided.

Author Response

Dear Editor,

Thank you very much for your time and effort in handling our manuscript. Please find the responses to reviewer 2, as follows:

Reviewer 2:

The manuscript agronomy-1717855 evaluates the rhizosphere-associated microbiome profile of agricultural lands located in three Egyptian major areas. The aim of the study was to identify sources of microorganisms to be used as biofertilizers. The authors studied microbial communities by 16S rDNA metabarcoding and applied bioinformatics and statistics to investigate samples and predict metagenome functions. The study is interesting and could add knowledge to the field. I would improve only minor aspects. See specific comments below.

Keywords: they should not match the words present in the title.

Response: Thanks for the comment, the keywords that exist in the title were changed to other ones.

Introduction: The introduction correctly places the study in context. I would improve the clarity of the purpose of the study and the working hypotheses being tested:

L86: Clarify the purpose of the "relevant bacteria" you searched for.

Response: Thank you for the suggestion, we have rephrased the working hypothesis and the objective part in the introduction of the revised version considering the raised comment.

L88: Add working hypotheses.

Response: The hypothesis was added and the text was rewritten for clarity.

Materials and Methods: The authors described in sufficient detail the methods used. However, I would briefly describe the methods and instruments used for the laboratory methods stated in line 114.

Response: We are sorry, unintended mistake ended up with this part missing in the text. We revised this part carefully and added the missing part.

Results: The results description is clear. Some sentences need a moderate English revision.

Response: We have re-read the text and asked a native colleague to check our manuscript.

Discussion: The authors correctly discussed the results from the perspective of previous studies in the broadest context possible.

Response: We are glad for the comment.

Conclusions: The section is appropriate as the manuscript contains many elements. However, since the study was explorative, the authors should mention how they will continue future studies based on the findings obtained. I would also underline that the functional analysis carried out is only predictive. Some statements on the importance of an RNA-sequencing analysis to study the activity of the rhizosphere microbiome must be provided.

Response: Thank you for the suggested modifications. We have shortened the conclusion part and included the RNAseq statement as a further analysis that would be needed to complete the functional overview of the microbial community in the rhizosphere and also serves as a validator for the predicted function that has been reported in our study.

We are very grateful for all the effort and time dedicated to our manuscript,

Best regards

M Magdy

Round 2

Reviewer 1 Report

The revised version of the manuscript "Rhizosphere-Associated Microbiome Profile of Agriculture Reclaimed Lands in Egypt” is much improved. It is therefore now ready for publication. 

I only request a careful revision of spelling and punctuation throughout the manuscript, after accepting all revisions made. By way of example:

Line 36: Please, delete "." after [3].

Line 92: Please, delete "." after "source".